# Two Ferulic Acid Derivatives Inhibit Neuroinflammatory Response in Human HMC3 Microglial Cells via NF-*κ*B Signaling Pathway

**DOI:** 10.3390/molecules28052080

**Published:** 2023-02-22

**Authors:** Pei-Lin Li, Xiao-Xue Zhai, Jun Wang, Xiang Zhu, Lin Zhao, Shuang You, Chun-Yan Sang, Jun-Li Yang

**Affiliations:** 1CAS Key Laboratory of Chemistry of Northwestern Plant Resources and Key Laboratory for Natural Medicine of Gansu Province, Lanzhou Institute of Chemical Physics, Chinese Academy of Sciences (CAS), Lanzhou 730000, China; 2University of Chinese Academy of Sciences, Beijing 100049, China; 3College of Chemical Engineering, Northwest Minzu University, Lanzhou 730030, China; 4Shandong Laboratory of Yantai Advanced Materials and Green Manufacturing, Yantai 264000, China; 5Yantai Zhongke Research Institute of Advanced Materials and Green Chemical Engineering, Yantai 264010, China; 6Yaomazi Food Co., Ltd., Hongya 620360, China

**Keywords:** *Zanthoxylum armatum*, neuroinflammation, ferulic acid, HMC3, NF-*κ*B signaling pathway

## Abstract

Various physiological and pathological changes are related to the occurrence and development of neurodegenerative diseases. Neuroinflammation is a major trigger and exacerbation of neurodegenerative diseases. One of the main symptoms of neuritis is the activation of microglia. Thus, to alleviate the occurrence of neuroinflammatory diseases, an important method is to inhibit the abnormal activation of microglia. This research evaluated the inhibitory effect of trans-ferulic acid (TJZ-1) and methyl ferulate (TJZ-2), isolated from *Zanthoxylum armatum*, on neuroinflammation, by establishing the human HMC3 microglial cell neuroinflammation model induced by lipopolysaccharide (LPS). The results showed both compounds significantly inhibited the production and expression of nitric oxide (NO), tumor necrosis factor-*α* (TNF-*α*), and interleukin-1*β* (IL-1*β*) contents, and increased the level of anti-inflammatory factor *β*-endorphin (*β*-EP). Furthermore, TJZ-1 and TJZ-2 can inhibit LPS-induced activation of nuclear factor kappa B (NF-*κ*B). It was found that of two ferulic acid derivatives, both had anti-neuroinflammatory effects by inhibiting the NF-*κ*B signaling pathway and regulating the release of inflammatory mediators, such as NO, TNF-*α*, IL-1*β*, and *β*-EP. This is the first report that demonstrates that TJZ-1 and TJZ-2 had inhibitory effects on LPS-induced neuroinflammation in human HMC3 microglial cells, which indicates that two ferulic acid derivates from *Z. armatum* could be used as potential anti-neuroinflammatory agents.

## 1. Introduction

Neuroinflammation is an adaptive immune response by various immune cells (such as microglia and astrocytes) to stimuli (such as infection, trauma, stress and ischaemia) to the body through the release of inflammatory mediators (such as neurotoxic substances and inflammatory cytokines) [1]. The development of neuroinflammation first manifests itself through microglial cell activation [2]. Upon activation, the morphology of microglial cells changes, including cell proliferation, cell apoptosis, hypertrophy of the cell soma and the release by various inflammatory factors and endotoxins. In recent years, it was found that neuroinflammation is associated with the development of neurodegenerative diseases, including Parkinson’s disease (PD), Alzheimer’s disease (AD), Amyotrophic lateral sclerosis (ALS), and Huntington’s disease (HD) [3]. This is because the activation of microglial cells usually leads to the accumulation of abnormal proteins, which, in turn, trigger an inflammatory response [4]. The neurodegenerative diseases are a serious concern in the field of public health because their incurability and the increased number of cases each year [3]. In addition, the vast majority of currently available treatment options only reduce disease-related symptoms, and do not prevent or suppress the progression of the disease.

Microglia are the main reservoir of immune cells within the brain and spinal cord, and play an important role in human health as the resident immune cells of the central nervous system [5]. Microglial cells have been shown to be involved in brain development, neuromodulation, synaptic plasticity, and to contribute to learning and memory processing [5]. When neuroinflammation happens, it is usually activated by lipopolysaccharide (LPS), interferon (IFN)-γ and *β*-amyloid, in order to resist external disturbances and keep homeostatic balance [6,7]. In parallel, microglial cells express membrane receptors and molecules, such as receptors for advanced glycation end-products (RAGE), clusters of differentiation-36 (CD-36), scavenger receptors (SRs), Fc receptors, complement receptors (CRs), and Toll-like receptors (TLRs), to detect aggressive factors and imbalances in homeostasis [8]. Upon activation, microglial cells are converted to the pro-inflammatory phenotype M1. During this process, the release of some inflammatory factors and neurotoxic substances, including tumor necrosis factor-*α* (TNF-*α*), interleukin-1*β* (IL-1*β*) and nitric oxide (NO), increase; while the levels of some anti-inflammatory factors, such as *β*-endorphin (*β*-EP), decrease, all as a sign that the body is fighting external disturbances to restore homeostatic balance [9]. Therefore, regulating the release of related inflammatory factors and neurotoxic substances may serve as a therapeutic entry point for neuroinflammation [10].

Trans-ferulic acid (TJZ-1) and methyl ferulate (TJZ-2) are two ferulic acid derivatives isolated from *Z. armatum* by our research group, and are naturally active substances with anti-hypoxic effects [11]. Many studies have shown that ferulic acid has multiple biological activities, including oxidative stress, inflammation, fibrosis, vascular endothelial injury and apoptosis [12]. Ferulic acid and berberine also promote cell survival and healthy longevity through Sirt1 and AMPK, while inhibiting NF-*κ*B-driven inflammation [13]. Meanwhile, it is reported that ferulic acid has potential anti-inflammatory effects by inhibiting LPS-induced inflammation via suppressing NK-*κ*B and MAPK signaling pathways [14]. Another study shows ferulic acid amide derivatives with antioxidant and anti-inflammatory activities [15]. Furthermore, it was found that ferulic acid improved placental inflammation and cell apoptosis in rats with pre-eclampsia [16]. Kim et al. [17] demonstrated that long-term administration of ferulic acid had inhibitory effects on microglial activation induced by intracerebroventricular injection of bb-amyloid peptide (1–42) in mice. Cheng et al. [18] found that ferulic acid can provide neuroprotection against oxidative stress-related apoptosis by inhibiting the expression of ICAM-1 mRNA after cerebral ischemia/reperfusion injury in rats. Sultana et al. [19] studied ferulic acid protection against amyloid *β*-peptide(1–42)-induced oxidative stress and neurotoxicity. Jin et al. [20] concluded that sodium ferulate prevented amyloid-beta-induced neurotoxicity through the suppression of p38 MAPK and upregulation of ERK-1/2 and Akt/protein kinase B in rat hippocampus. Herein, our research evaluated and analyzed whether two ferulic acid derivatives, TJZ-1 and TJZ-2, isolated from *Z. armatum*, with anti-hypoxic effects, could inhibit neuroinflammation in human HMC3 microglial cells. In this process, cell apoptosis is similarly involved in the development of neuroinflammation. In addition, this research indicated that TJZ-1 and TJZ-2 could inhibit LPS-induced neuroinflammation in human HMC3 microglial cells by inhibiting TLR4 activation. Furthermore, the activation of downstream target NF-*κ*B was inhibited, and inflammatory factors and neurotoxic substances were regulated. In conclusion, two ferulic acid derivatives, TJZ-1 and TJZ-2, isolated from *Z. armatum*, are expected to be effective candidates for inhibiting neuroinflammation.

## 2. Results

### 2.1. Effects of TJZ-1 and TJZ-2 on the Viability of HMC3 Cells

The chemical structures of two ferulic acid derivatives, TJZ-1 and TJZ-2, isolated from *Z. armatum*, are shown in Figure 1A. To evaluate the cytotoxicity of TJZ-1 and TJZ-2 on the viability of human HMC3 microglial cells, cells were pretreated with various concentrations, at 1, 5, 10 and 20 μM for 24 h. As shown in Figure 1B,C, TJZ-1 and TJZ-2 had no significant effect of human HMC3 microglial cell viability in the concentration range of 1, 5 and 10 µM. Therefore, the non-cytotoxic concentrations of 1 and 10 µM were used in the following experiments. Moreover, TJZ-1 and TJZ-2 combined with LPS treatment, respectively, were used to evaluate cell viability of human HMC3 microglial cells. Herein, celecoxib (10 μM), a nonsteroidal anti-inflammatory drug (NSAID) and cyclooxygenase-2 (COX-2) inhibitor, was used as an active control drug. Neuroinflammation of human HMC3 microglial cells was pretreated with LPS (1 μg/mL) for 24 h. The results showed that TJZ-1 and TJZ-2, respectively, combined with human HMC3 microglial cells, had no significant cytotoxicity compared with the control group (untreated with LPS, celecoxib, TJZ-1 or TJZ-2) (*p* > 0.05) (Figure 1D,E). From the above results, it can be seen that treatment with TJZ-1 and TJZ-2, respectively, at the concentrations of 1 and 10 µM, had no cytotoxic effects on human HMC3 microglial cells. Furthermore, combined treatment with LPS and celecoxib did not show cytotoxicity either.

### 2.2. Effects of TJZ-1 and TJZ-2 on LPS-Induced Neuroinflammation

The anti-inflammatory effect of TJZ-1 and TJZ-2 was evaluated by measuring NO production. This is the main inflammatory mediator in LPS-induced activation of human microglial cells. The release of NO in the culture supernatant was measured indirectly by the Griess method. Celecoxib (10 μM) was used as a positive drug. After LPS treatment, human HMC3 microglial cells significantly produced NO as compared to control cells (untreated with LPS, celecoxib, TJZ-1 or TJZ-2) (*p* < 0.0001). This suggests that neuroinflammation occurred after LPS stimulation of human HMC3 microglial cells. As shown in Figure 2A, TJZ-1 significantly reduced NO production at 1 and 10 μM. In addition, TJZ-1 (10 μM) activity was comparable to that of celecoxib drug (10 μM). However, TJZ-2 (1 μM) had no obvious inhibitory effects on NO release, while TJZ-2 (10 μM) significantly reduced LPS-induced NO production in human HMC3 microglial cells (Figure 2B).

In addition, anti-inflammatory activity of TJZ-1 and TJZ-2 on LPS-induced human HMC3 microglial cells was measured via the determination of TNF-*α*, IL-1*β* and *β*-EP levels by the ELISA method. The results showed that LPS increased the levels of TNF-*α* and IL-1*β*, and decreased the levels of *β*-EP extremely significantly (*p* < 0.0001) compared to the control group, indicating that microglial cellular inflammation was successfully induced by LPS treatment. Meanwhile, the positive drug, celecoxib, significantly (*p* < 0.0001) resisted LPS-induced microglial activation compared to the control group. According to measurements, TJZ-1 and TJZ-2 (1–10 μM), respectively, inhibited the production of TNF-*α* (Figure 2C). However, TJZ-1 inhibited IL-1*β* production only at 10 μM, while *β*-EP was inhibited under TJZ-2 (10 μM) treatment (Figure 2D,E). The results suggest that the anti-inflammatory effect of TJZ-1 and TJZ-2 on TNF-*α* levels may be more critical in the process.

### 2.3. Effects of TJZ-1 and TJZ-2 on LPS-Induced HMC3 Cell Late Apoptosis

To determine the effects of TJZ-1 and TJZ-2 on late apoptosis during LPS-induced inflammation in microglia, cell apoptosis was detected by Annexin V/PI co-staining. The control group was not subjected to FITC and PI staining (Figure 3A). As shown by the results, the flow cytometry assay revealed that the late apoptosis rate of human HMC3 microglial cells increased from 1.45% to 4.92% (*p* < 0.0001) (Figure 3B), and the positive group reversed the late apoptosis of cells, with the apoptosis rate being reduced to 2.49% (*p* < 0.0001) (Figure 3C). No significant improvement of apoptosis was obtained by TJZ-1 (Figure 3D) and TJZ-2 (Figure 3F) at low concentration (1 μM). On the other hand, 10 μM TJZ-1 and TJZ-2 reduced the late apoptosis rate of cells to 2.65% (*p* < 0.0001) (Figure 3E) and 2.71% (*p* < 0.0001) (Figure 3G), respectively, close to celecoxib. These results suggest that TJZ-1 and TJZ-2 had greater effects on late apoptosis of human HMC3 microglial cells at a high concentration (10 μM) (Figure 3H).

### 2.4. TJZ-1 and TJZ-2 Inhibit LPS-Induced HMC3 Activation via the NF-κB Signaling Pathway

Furthermore, the molecular mechanism of TJZ-1 and TJZ-2 inhibited the release of inflammatory cytokines and NO, as well as enhanced late cell apoptosis, as demonstrated by the Western blot assay based on human HMC3 microglial cells. The NF-*κ*B signaling pathway played an important role in the inflammatory process, and the NF-*κ*B signaling pathway regulated the pro-inflammatory effect by phosphorylating p65 and p65 nuclear ectopic in the microglial activation state [21]. In the inflammatory response of microglial cells, the NF-*κ*B signaling pathway was activated and the IKK complex was degraded. Subsequently, I*κ*B*α* was phosphorylated, reducing the nuclear translocation of the p65 complex [22]. In Figure 4A, the results of the Western blot showed that in HMC3 cells activated by LPS, the expression of IKK and p65 increased. Meanwhile, the expression of I*κ*B*α* decreased, and the positive group decreased the ratio of phosphorylated proteins and increased the content of I*κ*B*α* (Figure 4A). As shown in Figure 4B, TJZ-1 (1 and 10 μM) significantly inhibited the phosphorylation of IKK. However, TJZ-2 showed no inhibition of the phosphorylation of IKK (Figure 4B). The relative expression of I*κ*B*α* in Figure 4C showed that the degradation of I*κ*B*α* protein was not inhibited by treatment with TJZ-1 (1 and 10 μM) and TJZ-2 (1 and 10 μM). Moreover, p65 phosphorylation was significantly inhibited by treatment with 10 μM of TJZ-1 and TJZ-2, respectively (Figure 4D). These results suggest that TJZ-1 and TJZ-2 inhibit LPS-induced HMC3 activation via the NF-*κ*B signaling pathway.

## 3. Discussion

Natural products and natural product-active substances are valuable and significant for drug discovery and development, with 49.2% of the 1,881 drugs developed between 1981 and 2019 being developed directly from natural products or natural-product derivatives [23]. *Z. armatum* belongs to the family the Rutaceae and is widely distributed in the China and India areas. *Z. armatum* is a conventional medical and edible plant: its ripened and dried pericarps are often used as a condiment with a distinctive peppery flavor. In traditional medicine, *Z. armatum* is used for curing toothache, diabetes, eczema, diarrhea, inflammation and cancer [24]. The branches of *Z. armatum* are also often used folklorically as an antiseptic and insect repellent, and their processed products are also commonly used in human life as oral hygiene preparations and scabies lotions [25]. In this study, we propose that two natural compounds, the ferulic acid derivatives TJZ-1 and TJZ-2, derived from *Z. armatum*, could inhibit LPS-induced neuroinflammation in human HMC3 microglial cells.

In regulating the microenvironment of the brain, human HMC3 microglial cells play an important role and are often in an inactive state. Thus, when in an activated state, human HMC3 microglial cells normally secrete neurotoxic substances and inflammatory factors in response to changes their surroundings. Once these neurotoxic substances and inflammatory factors are released, they trigger an inflammatory cascade response that impairs neurological function. Although NO is an important regulatory element for physiological function, it is one of the most important inflammatory factors that causes neuronal cell damage and death. The inhibition of NO production is an effective and direct treatment for neuroinflammation. Here, we found that TJZ-1 and TJZ-2 inhibited the activation of human HMC3 microglial cells and alleviated late cell apoptosis. TNF-*α*, IL-1*β*, and *β*-EP play a key role in neurodegenerative diseases, mainly by regulating the release of inflammatory factors from glial cells, and inducing the expression of signaling-related transcription factors. Moreover, the results showed that TJZ-1 and TJZ-2 regulated the production of TNF-*α*, IL-1*β*, and *β*-EP.

Cell membranes in the central nervous system widely express TLR4. They are stimulated by endotoxins and activate the NF-*κ*B signaling pathway in microglia by endotoxin release, and regulate the production of inflammatory mediators [26]. The NF-*κ*B signaling pathway is the classical pathway of microglia-mediated neuroinflammation exposed to a variety of stimuli [27]. Under regular circumstances, microglia are in an inactivated state [28]. NF-*κ*B, associates with its inhibitor, I*κ*B, to constitute a complex that exists in the cytoplasm in an inactive state. In the activated state, I*κ*B*α* is phosphorylated by the I*κ*B kinase, which leads to the degradation of I*κ*B*α* [29]. When NF-*κ*B is phosphorylated and the IKK complex is degraded, RelA/p65 translocates to the nucleus and further regulates inflammation-associated gene expression. Ultimately, the NF-*κ*B signaling pathway regulates inflammation, cell apoptosis and neuroinflammation [30]. Feng et al. studied the inhibitory effects of ferulic acid on the inflammatory response in BV2 microglial cells induced by lipopolysaccharides [31]. In this study, human HMC3 microglial cells were used to evaluate ferulic acid and its derivates on their anti-neuroinflammation properties. In accordance with the findings of Feng et al., ferulic acid and its derivates inhibited the production of NO and IL-1*β* in LPS-induced human HMC3 microglial cells. Additionally, Chan et al. and Becher et al. found that LPS activates human-derived microglia, which, in turn, activates TLR4, causing a high expression of TNF-*α* and IL-6 [32]. Our study also found that TJZ-1 and TJZ-2 reduced TNF-*α* contents [32,33]. Baek et al. studied how BET attenuates the inflammatory response and cell migration in the human microglial HMC3 cell line [34]. Overall, the results revealed that TJZ-1 and TJZ-2 inhibit LPS-induced neuroinflammatory response in human HMC3 microglial cells via the NF-*κ*B signaling pathway. TJZ-1 and TJZ-2 regulated the expression of inflammatory mediators, such as NO, TNF-*α*, IL-1*β*, and *β*-EP. In addition, TJZ-1 and TJZ-2 had a greater effect on the late apoptosis of human HMC3 microglial cells at 10 μM. In the present study, TJZ-1 and TJZ-2 inhibited the activation of IKK, which, in turn, inhibited the degradation of I*κ*B*α*. Therefore, the nuclear translocation of RelA/p65 occurred, which alleviated the development of neuroinflammation. Additionally, TJZ-2 inhibited the degradation of IKK in a distinct manner to TJZ-2, however, further studies are needed. We speculate that this difference is likely to be due to a change in the carboxyl site.

In conclusion, this paper investigated the neuroinflammatory protective effects of the two ferulic acid derivatives TJZ-1 and TJZ-2 isolated from *Z. armatum*, in human HMC3 microglial cells, and revealed their anti-neuroinflammatory mechanism via the NF-*κ*B signaling pathway (Figure 5). It was indicated that different action of TJZ-1 and TJZ-2 in the role of NF-*κ*B signaling pathway may be due to the carboxyl site. The protective role of TJZ-1 and TJZ-2 against neuroinflammation in vivo awaits further study. TJZ-1 and TJZ-2 may have the potential opportunity to slow down neurodegenerative diseases.

## 4. Materials and Methods

### 4.1. Materials

Two ferulic acid derivatives, trans-ferulic acid (TJZ-1) and methyl ferulate (TJZ-2), were obtained by our research group from *Z. armatum*, and the relevant spectral data were in the reference [11]. The HPLC purity level was > 98%. Other reagents materials included lipopolysaccharide (LPS) (Solarbio, Beijing, China), celecoxib (CSPC Pharmaceutical Group Limited, Shijiazhuang, China), MTT (3-(4,5-dimethyl-2-thiazolyl)-2,5-diphenyl-2-*H*-tetrazolium bromide, Kum-amoto, Japan), and DMSO (Dimethyl sulfoxide) (Sigma, St Louis, MO, USA).

### 4.2. Cell Culture and Grouping

Human microglial cell lines (HMC3) were procured from Shanghai FHS Biotechnology Co., Ltd. (Shanghai, China). Human HMC3 microglial cells were cultured in Dul-becco modified Eagle’s medium (DMEM, Gibco, Carlsbad, CA, USA) comprising 10% fetal bovine serum (FBS, Sijiqing, Hangzhou, China), in an atmosphere of 5% CO_2_ at 37 ℃ in an incubator. For in vitro assays, two ferulic acid derivatives, TJZ-1 and TJZ-2, were dissolved in DMSO and diluted in the relevant culture media to a final DMSO concentration of 0.1% (*ν*/*ν*^−1^). All experiments used cells in the logarithmic growth phase after passaging by cell trypsin-EDTA (Gibco, Carlsbad, CA, USA). Human HMC3 microglial cells were stimulated with LPS (1 μg/mL) (Solarbio, Beijing, China) for 24 h in advance. Celecoxib acted as a positive control drug [35]. The experimental groups were the control group (without any treatment) and the LPS treatment group ((1 μg/mL), celecoxib (10 μM), TJZ-1 treatment (1 μM), TJZ-1 treatment (10 μM), TJZ-2 treatment (1 μM), TJZ-2 treatment (10 μM)).

### 4.3. Cell Viability Assay

An MTT (Kum-amoto, Japan) assay was showed to detect cell viability [36]. Human HMC3 microglial cells (3 × 10^3^/well) were seeded into 96-well plates. Each group was treated for 24 h overnight. The old medium was thrown away and 100 µL of medium containing 10 µL of MTT solution was added to the 96-well plates. Then, the plate was incubated for 4 h and 100 µL DMSO was added to dissolve crystals completely on the shaker. The optical density (OD) was then measured at 570 nm using a microplate reader (Rayto, Shenzhen, China). The cell viability (%) = (OD_570_ of treated group − OD_570_ of blank control group)/(OD_570_ of untreated group − OD_570_ of blank control group) × 100%.

### 4.4. Nitric Oxide Assay

The amount of NO produced by human HMC3 microglial cells was measured indirectly by the Griess reagent method [37]. Human HMC3 microglial cells in the logarithmic growth phase were stimulated in advance with LPS (1 μg/mL) for 24 h, at a cell density of 1 × 10^5^/mL in 96-well plates, then human HMC3 microglial cells were treated with TJZ-1 and TJZ-2, respectively, at concentrations of 1 μM and 10 μM, for 24 h. Celecoxib (10 μM) acted as an active control. Then, the cell supernatants were collected and mixed with assay reagents (Beyotime, Shanghai, China) to detect sodium nitrite levels at 562 nm, and the concentration of sodium nitrite was determined from a standard curve of known sodium nitrite concentrations.

### 4.5. Measurement of Inflammatory Factor Content

The ELISA method was used to measure the release of TNF-*α*, IL-1*β*, and *β*-EP (Mei Biao Biological Technology, Jiangsu, China) in human HMC3 microglial cells induced by LPS. At 37 °C and a cell density of 1 × 10^5^/mL, human HMC3 microglial cells were treated with TJZ-1 and TJZ-2 at concentrations of 1 μM and 10 μM, for 24 h, respectively. In a 96-well plate, the cells (10 × 10^5^/mL) were pretreated with LPS (1 µg/mL) for 24 h, then treated with TJZ-1 and TJZ-2, respectively, for 24 h, and the cell culture supernatant was collected. Celecoxib (10 μM) acted as an active control. The cell supernatant was centrifuged at 120× *g* for 10 min and then detected by a double-antibody sandwich ELISA method. The contents of inflammatory factors TNF-*α*, IL-1*β*, and *β*-EP were quantified using ELISA kits (Mei Biao Biological Technology, Jiangsu, China) according to the manufacturer’s instructions [38].

### 4.6. Apoptosis Assay

Annexin V/PI co-staining method was used to detect the degree of cell apoptosis. Human HMC3 microglial cells (1 × 10^5^/mL) were placed in 6-well plates and stimulated with LPS (1 μg/mL) for 24 h. The cells were then treated with TJZ-1 and TJZ-2, respectively, at concentrations of 1 μM and 10 μM, for 24 h. Celecoxib (10 μM) acted as an active control. Then, apoptosis assays were performed according to the protocol of the apoptosis kits (BD Biosciences, San Jose, CA, USA). Human HMC3 microglial cells were washed three times using serum-free medium, and 100 µL of binding buffer was added and dyed in the dark. Finally, a flow cytometer (Agilent NovoCyte, Palo Alto, CA, USA) was used for analysis at an excitation wavelength of 488 nm and an emission wavelength of 525 nm [39].

### 4.7. Western Blot

Human HMC3 microglial cells were stimulated with LPS (1 μg/mL) for 24 h, then cells were treated with 1 μM and 10 μM samples for 24 h. Cells were next lysed with Western blot cell lysis buffer (BioSharp, Hefei, China), added to the sample buffer, boiled and stored at −80 °C. Cell lysates were separated by SDS-PAGE (BioSharp, Hefei, China) and electrophoresed on PVDF membranes. The antibodies against IKK (#61294s, Cell Signaling Technology, Danvers, MA, USA), p-IKK (#2697s, Cell Signaling Technology, Danvers, MA, USA), I*κ*B*α* (#4812s, Cell Signaling Technology, Danvers, MA, USA), p65 (#8242s, Cell Signaling Technology, Danvers, MA, USA), p-p65 (#3033s, Cell Signaling Technology, Danvers, MA, USA), and *β*-actin (TA-09, zsbio, Beijing, China) were closed with 5% skimmed milk powder in a shaker for 2 h, and washed three times with a TBST buffer at 4 °C, diluted with anti-rabbit (ZB-2301, zsbio, Beijing, China) or anti-mouse (ZB-2305, zsbio, Beijing, China) lgG horseradish enzyme in TBST (1:8000), and incubated at room temperature for 2 h. After treatment with chemiluminescence agent (BioSharp, Hefei, China), protein bands were detected by a chemiluminescence instrument (SAGECREATION, Beijing, China) and quantified by Image J 1.52a (National Institutes of Health, Bethesda, MD, USA) [40].

### 4.8. Statistical Analyses

Data are expressed as the mean ± SDs based on at least three independent experiments. Statistical analysis was performed using the GraphPad Prism 8.0.2 software. Data from groups were analyzed by a one-way analysis of variance (ANOVA), followed by a Tukey’s tests. A *p* value of less than 0.05 was considered a significant difference [41].

## Figures and Tables

**Figure 1 molecules-28-02080-f001:**
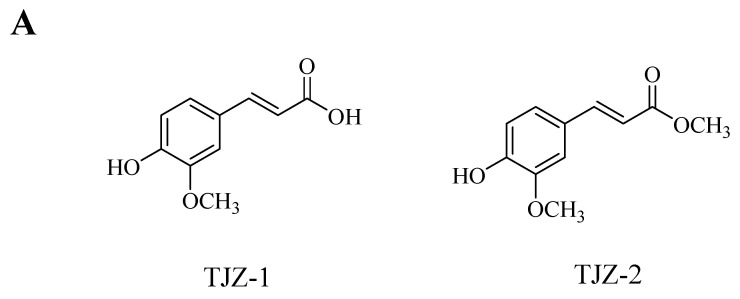
Effect of TJZ-1 and TJZ-2 on the viability of human HMC3 microglial cells. (**A**) The chemical structure of TJZ-1 and TJZ-2. (**B**,**C**) Human HMC3 microglial cells were treated with 0, 1, 5, 10 and 20 μM TJZ-1 and TJZ-2 for 24 h. (**D**,**E**) The cells were treated with LPS (1 μg/mL), and then treated with 1 and 10 μM TJZ-1 and TJZ-2, respectively, for 24 h. Celecoxib (10 μM) was used as a positive drug. The control group was untreated with LPS, celecoxib, TJZ-1 or TJZ-2. The cell survival rate was determined by an MTT assay. “ns” was not statistically significant. **** *p* < 0.0001 compared with control group. Data are expressed as the mean ± SD (*n* = 3).

**Figure 2 molecules-28-02080-f002:**
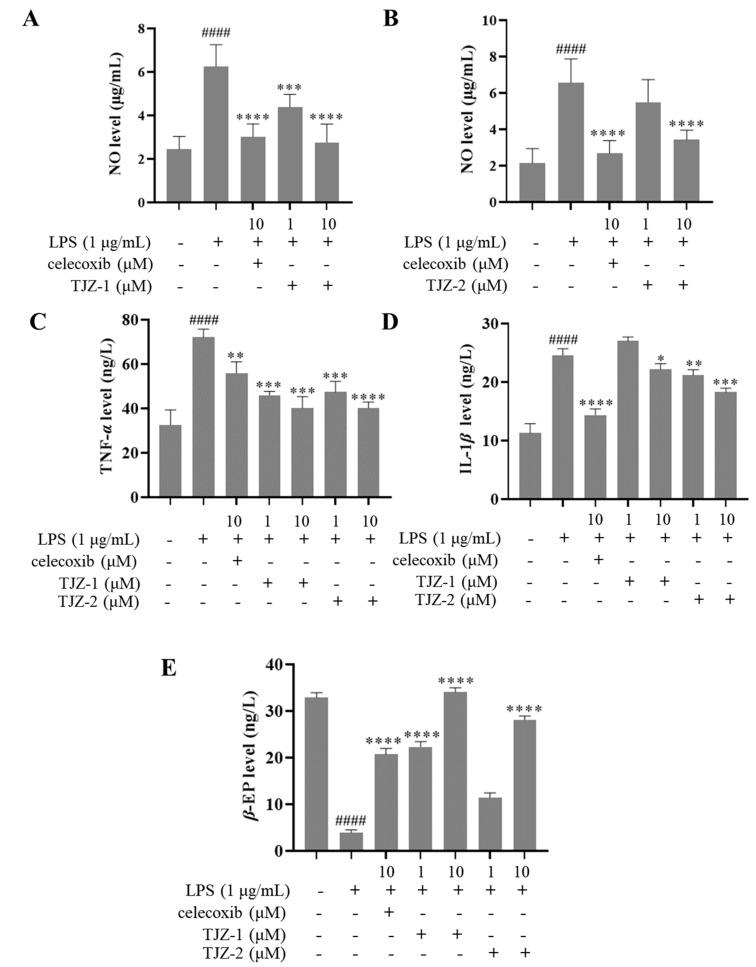
Effects of TJZ-1 and TJZ-2 on neuroinflammation induced by LPS in human HMC3 microglial cells. After stimulating the cells with LPS for 24 h, the supernatant was collected after 24 h of treatment with TJZ-1 and TJZ-2. Celecoxib (10 μM) was used as a positive drug. The control group was untreated with LPS, celecoxib, TJZ-1 or TJZ-2. (**A**,**B**) The content of NO. The production of (**C**) TNF-*α*; (**D**) IL-1*β*; (**E**) *β*-EP in human HMC3 microglial cells after LPS stimulation. #### *p* < 0.0001 compared with the control group. * *p* < 0.05, ** *p* < 0.01, *** *p* < 0.001, **** *p* < 0.0001 compared with the LPS-treated group. Data are expressed as the mean ± SD (*n* = 3).

**Figure 3 molecules-28-02080-f003:**
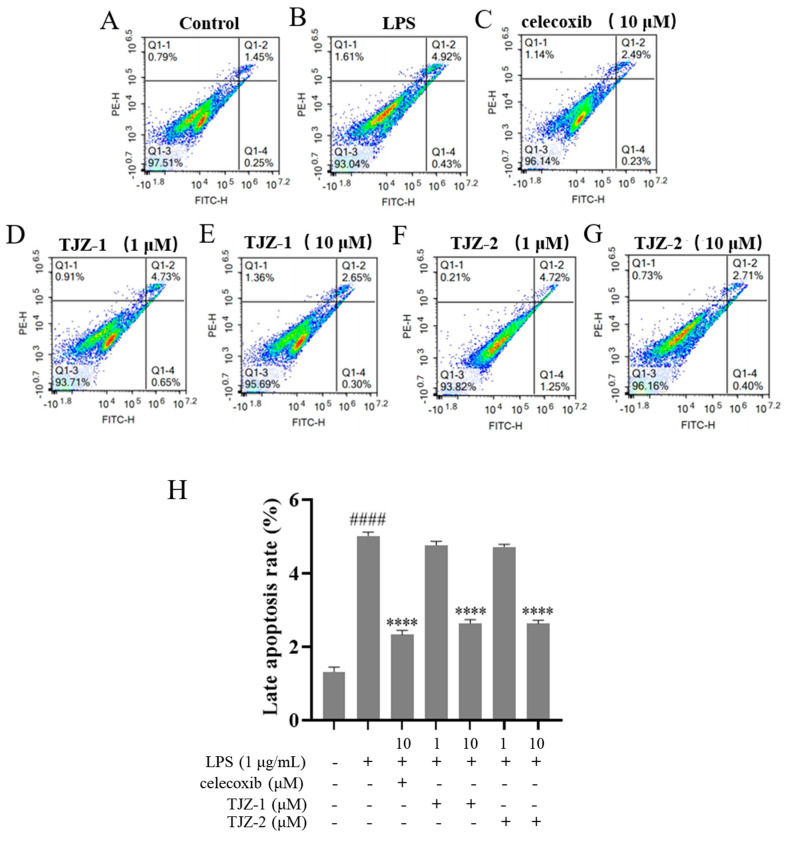
Effects of TJZ-1 and TJZ-2 on LPS-induced late apoptosis in human HMC3 microglial cells. (**A**–**G**) Human HMC3 microglial cells were treated with TJZ-1/TJZ-2 (1 and 10 μM) for 24 h, followed by treatment with LPS (1 μg/mL) for 24 h. Celecoxib (10 μM) as a positive control drug. The control group was untreated with LPS, celecoxib, TJZ-1 or TJZ-2. Annexin V/PI co-staining was used to detect the rate of late apoptosis. (**H**) Statistics on the rate of late apoptotic cell death. **** *p* < 0.0001 compared with the LPS-treated group; #### *p* < 0.0001 compared with the control group. Data are expressed as the mean ± SD (*n* = 3).

**Figure 4 molecules-28-02080-f004:**
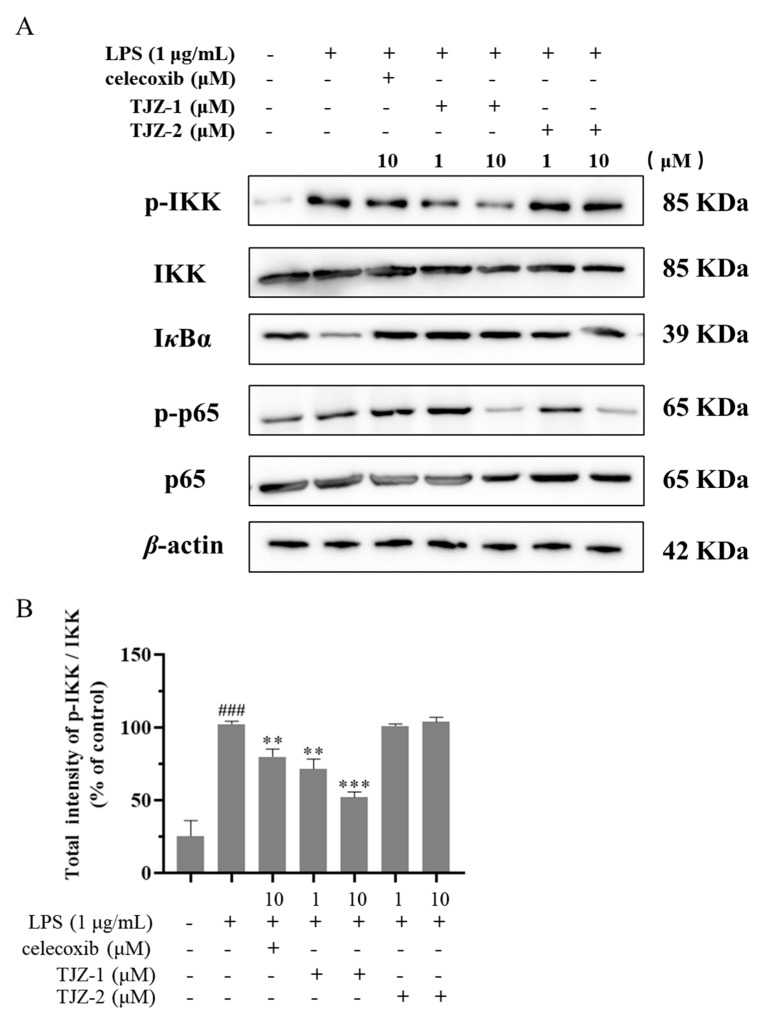
TJZ-1 and TJZ-2 inhibit LPS-induced human HMC3 microglial cell activation through the NF-*κ*B signaling pathway. Human HMC3 microglial cells were treated with LPS (1 μg/mL) for 24 h, followed by the treatment with TJZ-1 and TJZ-2 (1 and 10 μM), respectively, for 24 h. Celecoxib (10 μM) as a positive control drug. The control group was untreated with LPS, celecoxib, TJZ-1 or TJZ-2. (**A**) The effect of TJZ-1 and TJZ-2 on the protein expression of p-IKK, IKK, I*κ*B*α*, p-p65 and p65 was detected by a Western blot assay. (**B**–**D**) The relative protein expression of p-IKK/IKK, I*κ*B*α* and p-p65/p65 in the Western blot assay. * *p* < 0.05, ** *p* < 0.01; *** *p* < 0.001 compared with the LPS-treated group; ## *p* < 0.01, ### *p* < 0.001 compared with the control group.

**Figure 5 molecules-28-02080-f005:**
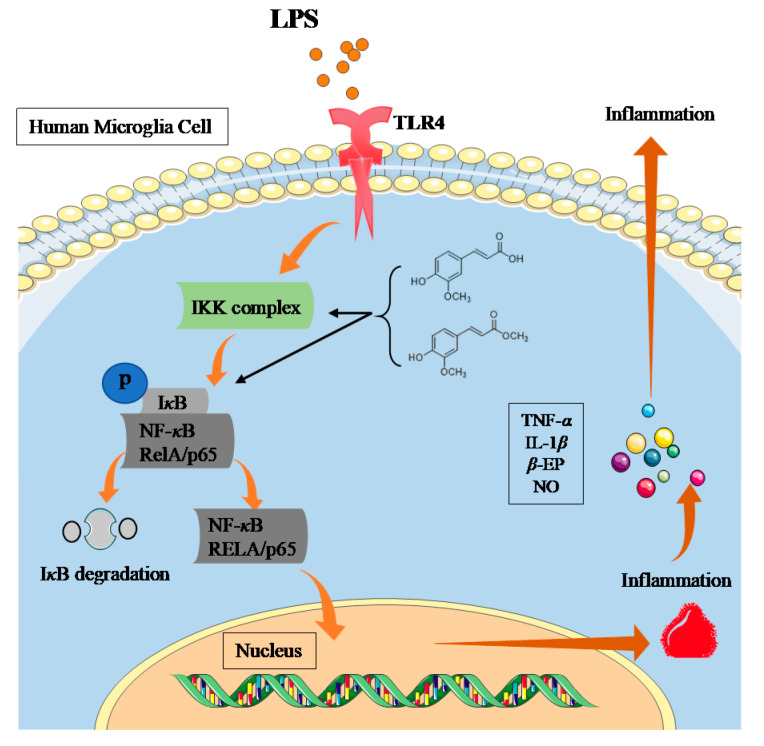
A schematic diagram of the potential molecular mechanism of the anti-neuroinflammatory effects of two ferulic acid derivatives, TJZ-1 and TJZ-2, isolated from *Z. armatum*, through the NF-*κ*B signaling pathway.

## Data Availability

Data are contained within the article.

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
