# Peer review of "Two Ferulic Acid Derivatives Inhibit Neuroinflammatory Response in Human HMC3 Microglial Cells via NF-κB Signaling Pathway"

_molecules, 2023, doi:10.3390/molecules28052080_

Round 1

Reviewer 1 Report

1. The title of this article seems a little long. Could the author use more concise sentences to express?

2. Some of the key words do not coincide with the main content of the manuscript, the author should choose more suitable ones.

3. For Introduction, it is suggested that the author provide more details about TJZ-1 and TJZ-2, such as what polarity are these substances separated.

4. For the results of 2.1, the author described the HMC3 cells were pre-treated with various concentrations at 1, 5, 10 and 20 μM for 24 h. In this process, why did the authors jump directly from 10 μM to 20 μM without adding a concentration of 15 μM ?

5. The author claimed TJZ-1 and TJZ-2 inhibit LPS induced HMC3 activation via the NF-κB signaling pathway. Why did the authors choose to study this signaling pathway? Has The author considered other signaling pathways?

6. For the Discussion, the author is almost describing the results of the experiment again here. Its necessary to have an in-depth discussion on the basis of the experimental results.

7. The number of references is insufficient, and the latest literatures are few.

Author Response

Dear reviewer,

Thanks very much for your careful reading and kind comments on the research and also thanks for providing this valuable opportunity to revise our manuscript. We extremely cherish this opportunity to revise. In response to your concerns and questions, we made the following responds point by point.

Point 1: The title of this article seems a little long. Could the author use more concise sentences to express?

Response 1: According to this comment, the title has been revised.

Point 2: Some of the key words do not coincide with the main content of the manuscript, the author should choose more suitable ones.

Response 2: According to this comment, the key words has been revised.

Point 3: For Introduction, it is suggested that the author provide more details about TJZ-1 and TJZ-2, such as what polarity are these substances separated.

Response 3: According to this comment, the article has been revised.

Point 4: For the results of 2.1, the author described the HMC3 cells were pre-treated with various concentrations at 1, 5, 10 and 20 μM for 24 h. In this process, why did the authors jump directly from 10 μM to 20 μM without adding a concentration of 15 μM ?

Response 4: When TJZ-1 and TJZ-2 were at low concentrations (1 μM), we used a high multiple (5-fold) to increase the concentration, after which we used a low multiple (2-fold) to increase the compound concentration to 20 μM to verify the activity.

Point 5: The author claimed TJZ-1 and TJZ-2 inhibit LPS induced HMC3 activation via the NF-κB signaling pathway. Why did the authors choose to study this signaling pathway? Has The author considered other signaling pathways?

Response 5: As you are concerned, NF-κB signaling pathway is a classic inflammatory signaling pathway. We will also consider other signaling pathways in the future research.

Point 6: For the Discussion, the author is almost describing the results of the experiment again here. It’s necessary to have an in-depth discussion on the basis of the experimental results.

Response 6: According to this comment, the Discussion has been revised.

Point 7: The number of references is insufficient, and the latest literatures are few.

Response 7: As you are concerned, we have added references including the latest literatures in our manuscript. 

Reviewer 2 Report

The manuscript entitled “Two Ferulic Acid Derivatives Inhibit Lipopolysaccharide-In-duced Neuroinflammatory Response in Human HMC3 Micro-glia Cells via NF-κB Signaling Pathway” submitted by Chun-Yan Sang to the Molecules MDPI journal is very well described.

It contains author list typo and is hard to see who is/are the corresponding author or authors:  “Pei-Lin Li 1, 2, Xiao-Xue Zhai 3, Jun Wang 4,5, Xiang Zhu 6, Lin Zhao 6, Shuang You 6, Chun-Yan Sang 1, *, Jun-Li Yang 1 and *”  

In order to find natural chemical compounds with anti-neurodegenerative therapeutic effects, trans-ferulic acid (TJZ-1) and methyl ferulate (TJZ-2) isolated from Zanthoxylum armatum were tested using the human HMC3 microglia cells neuroinflammation model induced by lipopoly-saccharide (LPS).  Based on the results both compounds inhibited the production and expression of nitric oxide (NO), tumor necrosis factor-α (TNF-α), interleukin-1β (IL-1β) contents and increase the level of anti-inflammatory factor β-endorphin (β-EP).  Ferulic acid derivates (TJZ-1 and TJZ-2) inhibited the LPS-induced neuroinflammatory besides both regulated the releasing of the inflammatory mediators such as NO, TNF-α, IL-1β, and β-EP in human HMC3 microglia cells, which indicated two ferulic acid derivates from Z. armatum could be used as potential anti-neuroinflammatory agents having a good potential opportunity to slow down neurodegenerative diseases.

Author Response

Dear reviewer,

Thanks very much for your decision and appreciate your comments. We extremely cherish this opportunity to revise. In response to your concerns and questions, we made the following responds.

Point 1: The manuscript entitled “Two Ferulic Acid Derivatives Inhibit Lipopolysaccharide-In-duced Neuroinflammatory Response in Human HMC3 Micro-glia Cells via NF-κB Signaling Pathway” submitted by Chun-Yan Sang to the Molecules MDPI journal is very well described.

Response 1: Dear reviewer, we wish to express our thankfulness for the kind remarks and positive comments on our paper.

Point 2: It contains author list typo and is hard to see who is/are the corresponding author or authors: “Pei-Lin Li 1, 2, Xiao-Xue Zhai 3, Jun Wang 4,5, Xiang Zhu 6, Lin Zhao 6, Shuang You 6, Chun-Yan Sang 1, *, Jun-Li Yang 1 and *” 

Response 2: Dear reviewer, thanks for your patience and suggestions for improvement. It has been revised.

Reviewer 3 Report

Pei-Lin and coworkers studied the effect of two derivatives of ferulic acid isolated from Zanthoxylum armatum in the activation of HCM3 cells induced by LPS. Both molecules TJZ-1 and TJZ-2 exhibited an inhibitory effect in the production of NO, and IL-1b. TNF-a. Moreover, the compounds modulate key molecules involved in the NFκB pathway, one of the main mechanisms involved in inflammation. The finding looks interesting. However, major changes must be addressed by the author before publication.

Major comments

  1. The authors must significantly improve the writing of the manuscript. The current version is hard to read. It contains several redundant and unclear ideas. This makes it difficult to analyze the quality of the science of the manuscript. Here are some examples: The anti-inflammatory effect of TJZ-1 and TJZ-2 was evaluated by detecting the release of NO. This is the major inflammatory mediators in human microglia cells induced by LPS. The release of NO in the culture supernatant was measured indirectly by Griess method, and the anti-inflammatory effects of TJZ-1 and TJZ-2 were evaluated [14]. Suggestion: The anti-inflammatory effect of TJZ-1 and TJZ-2 was evaluated by measuring NO production. This is the main inflammatory mediator in the LPS-induced activation of human microglia cells. The release of NO in the culture supernatant was measured indirectly by the Griess method [14].
  2. After LPS treatment, human HMC3 microglia cells individually significantly caused NO production compared to the control cells (p < 0.0001). It suggested that neuroinflammation occurred after LPS stimulation of human HMC3 microglia cells. Suggestion: After LPS treatment, human HMC3 microglia cells significantly produced NO as compared to control cells (p<0.0001). This suggests that neuroinflammation occurred after LPS stimulation of human HMC3 microglia cells.
  3. From the above results it can be seen that TJZ-1 and TJZ-2 respectively treatment at a concentration of 1-10 µM had no cytotoxicity to human HMC3 microglia cells. Furthermore, combined LPS treatment. Suggestion: From the above results, it can be seen that treatment with TJZ-1 and TJZ-2 respectively at the concentrations of 1 and 10 µM had no cytotoxicity to human HMC3 microglia cells. Furthermore, combined treatment with LPS and celecoxib did not show cytotoxicity either.

2. In the introduction the authors do not provide evidence that supports the protective effect of ferulic acid in neuroinflammation, please add id, or make your writing clearer. Thus, is unclear why they propose that two ferulic acid derivatives TJZ-1 and TJZ-2, could inhibit LPS-induced neuroinflammation in human HMC3 microglia cells. Add the evidence that supports it.

3. Why celecoxib was used as a positive control? This drug is well established by its anti-inflammatory properties mediated through the inhibition of COX activity, thus inhibiting PG production. However, it is not the best control to evaluate microglial activation. Why the authors did not use minocycline? A well-established drug that reduces microglial activation in vivo and in vitro?

3. In figure legend 1, what does mean blank control cells? Please replace it with a more specific term and add the label to the figure. Moreover, add which statistical test was used to compare statistical differences between groups.

4. Change the title of the figure legend 2. This figure is showing the Effect of TJZ-1 and TJZ-2 in LPS-induced NO production in human HMC3 glial cells, but not in neuroinflammation. This could be true if the authors place together figure 2 and 3. It is recommended to do this. The writing of the figure legend is confusing, please improve it. Moreover, add which statistical test was used to compare statistical differences between groups.

5. Improve de writing of the results of figure 5. The actual version is very hard to understand and is very difficult to follow. More order and sequence in the description of the results are recommended. Pay attention when mentioning in which panel of the figure these results are found. For example, the authors do not refer to the results expressed in figure 5A, i.e., they only mention the background or the text referring to the explanation they observed in the figure or at the observed in both this figure and the theory. But they did not include the number of the figure in the text. Another example is “Under TJZ-1 treatment, the phosphorylation of IKK and p65 (Figure 5D) and degradation of IκBα were significantly inhibited and the expression of IκBα under TJZ-1 treatment was significantly decreased (p < 0.001) (Figure 5C)” The redaction of this part is confusing. Moreover, is desirable for these results, to calculate and include the percentage of inhibition or increase of each marker.

6. In the discussion section:

a. the author wrote “Here research found that cell late apoptosis also played an important role in the development of neuroinflammation of human HMC3 microglia cells” However, the results did not support this conclusion. They just showed that TZJ1 and 2 had a greater effect on late apoptosis of human HMC3 microglia cells at 10 uM. Please, place in context your results.

b. The authors could use the results of the research of Feng, et. al. (2011) to discuss the differences or coincidences between the studies elaborated on murine and human microglial cells with ferulic acid and ferulic acids derivatives, respectively. To show the advance of knowledge on this topic (see the reference in the introduction section of this document).

C. Feng, et. al. (2011) hypothesizes that ferulic acid may inhibit neuroinflammation by inhibiting TLR4-related immune signaling pathways. Place in context this evidence with these results.

7. Section 4.1. Describe how and where plant material was collected, and how the procedure to isolate was. And how TZJ1 and TZJ2 were identified?

8. Section 4.8. Correct the statistical analysis description. A Student's t-test usually is not used followed by a one-way ANOVA.

9. I suggest including these two relevant references about the main topic and the method of this research (I explain why in this document of revision).

  • Feng, H.; Hua-Ming, D.; Miao-Miao, Z.; Fei, X.; Li, Y.; Zai-Jun, Z.; Ying, X.; Hong, N. Inhibitory effect of ferulic acid on inflammatory response in microglia induced by lipopolysaccharides. Zool. Res201132(3), 311-316. doi: 10.3724/SP.J.1141.2011.03311.

  • Baek, M.; Yoo, E.; Choi, H.I.; An, G.Y.; Chai, J.C.; Lee, Y.S.; Jung, K.H.; Chai, Y.G. The BET inhibitor attenuates the inflammatory response and cell migration in human microglial HMC3 cell line. Sci. Rep202111, 1-13. doi:10.1038/s41598-021-87828-1.

Minor comments

  1. Looks like the author list is incomplete. Please check if any author is missing
  2. Introduction section: In the context of the sentence “The neurodegenerative diseases and their complications are a serious concern in the field of public health because their incurability…” independence of patients looks incongruent. Is this term correct?
  3. Section 2.1 Replace “concentration was ” with concentrations 1 and 10 uM used in the …
  4. Result section. In “As shown in Figure 2A, TJZ-1 significantly reduced NO production in the concentration range of 1-10 μM” correct that TJZ-1 significantly reduced NO production at 1 and 10 μM, but not in that range of concentrations (the results did not support this conclusion).
  5. Delete (figure 6) from the conclusion section. This will be in the discussion section.
  6. Replace the schematic used to represent TLR4. It looks like an ionic channel, while TLR4 is a type I transmembrane protein composed of 22 extracellular leucine-rich repeats (LRRs), a transmembrane domain, and the Toll/IL-1 receptor domain (TIR domain) that is an essential for TLR signaling.

Author Response

Dear reviewer,

Thanks very much for taking your time to review this manuscript. I really appreciate all your comments and suggestions. We extremely cherish this opportunity to revise and have made revisions followed by point by point.

Point 1: The authors must significantly improve the writing of the manuscript. The current version is hard to read. It contains several redundant and unclear ideas. This makes it difficult to analyze the quality of the science of the manuscript. Here are some examples: The anti-inflammatory effect of TJZ-1 and TJZ-2 was evaluated by detecting the release of NO. This is the major inflammatory mediators in human microglia cells induced by LPS. The release of NO in the culture supernatant was measured indirectly by Griess method, and the anti-inflammatory effects of TJZ-1 and TJZ-2 were evaluated [14]. Suggestion: The anti-inflammatory effect of TJZ-1 and TJZ-2 was evaluated by measuring NO production. This is the main inflammatory mediator in the LPS-induced activation of human microglia cells. The release of NO in the culture supernatant was measured indirectly by the Griess method [14].

Response 1: According to the comment, the manuscript has been revised.

Point 2: After LPS treatment, human HMC3 microglia cells individually significantly caused NO production compared to the control cells (p < 0.0001). It suggested that neuroinflammation occurred after LPS stimulation of human HMC3 microglia cells. Suggestion: After LPS treatment, human HMC3 microglia cells significantly produced NO as compared to control cells (p < 0.0001). This suggests that neuroinflammation occurred after LPS stimulation of human HMC3 microglia cells.

Response 2: According to the comment, the manuscript has been revised.

Point 3: From the above results it can be seen that TJZ-1 and TJZ-2 respectively treatment at a concentration of 1-10 µM had no cytotoxicity to human HMC3 microglia cells. Furthermore, combined LPS treatment. Suggestion: From the above results, it can be seen that treatment with TJZ-1 and TJZ-2 respectively at the concentrations of 1 and 10 µM had no cytotoxicity to human HMC3 microglia cells. Furthermore, combined treatment with LPS and celecoxib did not show cytotoxicity either.

Response 3: According to the comment, the manuscript has been revised.

Point 4: In the introduction the authors do not provide evidence that supports the protective effect of ferulic acid in neuroinflammation, please add id, or make your writing clearer. Thus, is unclear why they propose that two ferulic acid derivatives TJZ-1 and TJZ-2, could inhibit LPS-induced neuroinflammation in human HMC3 microglia cells. Add the evidence that supports it.

Response 4: According to the comment, the relative statements have been added.

Point 5: Why celecoxib was used as a positive control? This drug is well established by its anti-inflammatory properties mediated through the inhibition of COX activity, thus inhibiting PG production. However, it is not the best control to evaluate microglial activation. Why the authors did not use minocycline? A well-established drug that reduces microglial activation in vivo and in vitro?

Response 5: According to the reviewers’ comments, minocycline is a tetracycline antibiotic and can bind to tRNA to achieve bacterial inhibition. Our experiments were based on LPS-induced microglia activation to investigate the neuroinflammatory protective effect of ferulic acid and derivatives. Celecoxib as NSAIDS anti-inflammatory drug is used clinically for the treatment of inflammatory pain. Hence, we have chosen celecoxib as a positive control drug.

Point 6: In figure legend 1, what does mean blank control cells? Please replace it with a more specific term and add the label to the figure. Moreover, add which statistical test was used to compare statistical differences between groups.

Response 6: Dear reviewer, we apologize for the poor language of our manuscript. “Blank control cells” has been revised to “control group”. Moreover, one-way ANOVA followed by Tukey’s tests was used to compare statistical differences between groups.

Point 7: Change the title of the figure legend 2. This figure is showing the Effect of TJZ-1 and TJZ-2 in LPS-induced NO production in human HMC3 glial cells, but not in neuroinflammation. This could be true if the authors place together figure 2 and 3. It is recommended to do this. The writing of the figure legend is confusing, please improve it. Moreover, add which statistical test was used to compare statistical differences between groups.

Response 7: According to the comment, the relative statements have been revised.

Point 8: Improve de writing of the results of figure 5. The actual version is very hard to understand and is very difficult to follow. More order and sequence in the description of the results are recommended. Pay attention when mentioning in which panel of the figure these results are found. For example, the authors do not refer to the results expressed in figure 5A, i.e., they only mention the background or the text referring to the explanation they observed in the figure or at the observed in both this figure and the theory. But they did not include the number of the figure in the text. Another example is “Under TJZ-1 treatment, the phosphorylation of IKK and p65 (Figure 5D) and degradation of IκBα were significantly inhibited and the expression of IκBα under TJZ-1 treatment was significantly decreased (p < 0.001) (Figure 5C)” The redaction of this part is confusing. Moreover, is desirable for these results, to calculate and include the percentage of inhibition or increase of each marker.

Response 8: According to the comment, the relative statements have been revised.

Point 9: In the discussion section: a. The author wrote “Here research found that cell late apoptosis also played an important role in the development of neuroinflammation of human HMC3 microglia cells” However, the results did not support this conclusion. They just showed that TZJ1 and 2 had a greater effect on late apoptosis of human HMC3 microglia cells at 10 uM. Please, place in context your results.

Response 9: Dear reviewers, thanks for your comments and they are really helpful. We revised our manuscript as follows: In addition, TJZ-1 and TJZ-2 had a greater effect on late apoptosis of human HMC3 microglia cells at 10 μM.

Point 10: In the discussion section: b. The authors could use the results of the research of Feng, et. al. (2011) to discuss the differences or coincidences between the studies elaborated on murine and human microglial cells with ferulic acid and ferulic acids derivatives, respectively. To show the advance of knowledge on this topic (see the reference in the introduction section of this document).

Response 10: Dear reviewer, we feel great thanks for your professional review. As you are concerned, we revised our manuscript. Feng, et. al (2011) studied inhibitory effect of ferulic acid on inflammatory response in BV2 microglia cells induced by lipopolysaccharides. They investigated the effect of ferulic acid on the cell viability of BV2 cells and levels of NO, PGE2, inflammatory factors. Morever, the research examined the effect of ferulic acid on the expression of iNOS and COX-2 proteins. It has been shown that ferulic acid inhibits microglial cell activation and suppression of neuroinflammation. This may be due to the fact that ferulic acid inhibits neuroinflammation by suppressing TLR4-related immune signaling pathways. Based on the study by Feng et al, here we used human-derived HMC3 microglia to study the protective effect of ferulic acid and its derivatives on neuroinflammation including effects on cell viability, and the role of inflammatory mediators. For the detection of IKK, IκBα, and p65 proteins in the NF-κB signaling pathway, it was found that ferulic acid and its derivatives inhibited the degradation of IκBα and thus increased the phosphorylation content of IKK and p65 proteins. This further helps us to understand the protective effect of ferulic acid compounds on neuroinflammation in the human brain.

Point 11: In the discussion section: c. Feng, et. al. (2011) hypothesizes that ferulic acid may inhibit neuroinflammation by inhibiting TLR4-related immune signaling pathways. Place in context this evidence with these results.

Response 11: Dear reviewer, thank your professional review work. As you suggested, we add these results in discussion section in our manuscript.

Point 12: Section 4.1. Describe how and where plant material was collected, and how the procedure to isolate was. And how TJZ-1 and TJZ-2 were identified?

Response 12: Dear reviewer, thank you so much for handling the review of our manuscript. Two ferulic acid derivatives trans-ferulic acid (TJZ-1) and methyl ferulate (TJZ-2) were obtained by our research group from Z. armatum. The relevant spectral data were in the reference [5].

Point 13: Section 4.8. Correct the statistical analysis description. A Student's t-test usually is not used followed by a one-way ANOVA.

Response 13: Dear Reviewers, thank you for your valuable comments. We have re-written this statistical analysis description as follows: Data from groups were analyzed by one-way analysis of variance (ANOVA) followed by Tukey’s tests.

Point 14: I suggest including these two relevant references about the main topic and the method of this research (I explain why in this document of revision).

Feng, H.; Hua-Ming, D.; Miao-Miao, Z.; Fei, X.; Li, Y.; Zai-Jun, Z.; Ying, X.; Hong, N. Inhibitory effect of ferulic acid on inflammatory response in microglia induced by lipopolysaccharides. Zool. Res. 2011, 32(3), 311-316. doi: 10.3724/SP.J.1141.2011.03311.

Baek, M.; Yoo, E.; Choi, H.I.; An, G.Y.; Chai, J.C.; Lee, Y.S.; Jung, K.H.; Chai, Y.G. The BET inhibitor attenuates the inflammatory response and cell migration in human microglial HMC3 cell line. Sci. Rep. 2021, 11, 1-13. Doi:10.1038/s41598-021-87828-1.

Response 14: We sincerely thank the reviewer for careful reading. We have added these references.

Point 15: Looks like the author list is incomplete. Please check if any author is missing.

Response 15: Dear reviewer, we feel great thanks for your careful review work. We have checked and confirmed that none of the author has been missed.

Point 16: Introduction section: In the context of the sentence “The neurodegenerative diseases and their complications are a serious concern in the field of public health because their incurability…” independence of patients looks incongruent. Is this term correct?

Response 16: According to the comment, the relative statements have been revised.

Point 17: Section 2.1 Replace “concentration was ” with concentrations 1 and 10 uM used in the …

Response 17: According to the comment, the relative statements have been revised.

Point 18: Result section. In “As shown in Figure 2A, TJZ-1 significantly reduced NO production in the concentration range of 1-10 μM” correct that TJZ-1 significantly reduced NO production at 1 and 10 μM, but not in that range of concentrations (the results did not support this conclusion).

Response 18: According to the comment, the relative statements have been revised.

Point 19: Delete (figure 6) from the conclusion section. This will be in the discussion section.

Response 19: Dear reviewer, thanks for your patience and suggestions for improvement. We have deleted Figure 6 from the conclusion section and added this in the discussion section.

Point 20: Replace the schematic used to represent TLR4. It looks like an ionic channel, while TLR4 is a type I transmembrane protein composed of 22 extracellular leucine-rich repeats (LRRs), a transmembrane domain, and the Toll/IL-1 receptor domain (TIR domain) that is an essential for TLR signaling.

Response 20: Dear reviewer, thanks for the professional comment. The corrections of TLR4 in the schematic diagram have been made.

References:

  1. Kim, H. S., Cho, J. Y., Kim, D. H., Yan, J. J., Lee, H. K., Suh, H. W., & Song, D. K. Inhibitory effects of long-term administration of ferulic acid on microglial activation induced by intracerebroventricular injection of beta-amyloid peptide (1-42) in mice. Biol Pharm Bull. 2004;27(1):120-121. doi:10.1248/bpb.27.120
  2. Cheng CY, Su SY, Tang NY, Ho TY, Chiang SY, Hsieh CL. Ferulic acid provides neuroprotection against oxidative stress-related apoptosis after cerebral ischemia/reperfusion injury by inhibiting ICAM-1 mRNA expression in rats. Brain Res. 2008;1209:136-150. doi:10.1016/j.brainres.2008.02.090
  3. Sultana, R., Ravagna, A., Mohmmad-Abdul, H., Calabrese, V. and Butterfield, D.A., Ferulic acid ethyl ester protects neurons against amyloid β- peptide(1–42)-induced oxidative stress and neurotoxicity: relationship to antioxidant activity. Journal of Neurochemistry. 2005 92: 749-758. doi:10.1111/j.1471-4159.2004.02899.x
  4. Jin, Y., Yan, E. Z., Fan, Y., Zong, Z. H., Qi, Z. M., & Li, Z.. Sodium ferulate prevents amyloid-beta-induced neurotoxicity through suppression of p38 MAPK and upregulation of ERK-1/2 and Akt/protein kinase B in rat hippocampus. Acta pharmacologica Sinica, 2005;26(8),943–951. doi:10.1111/j.1745-7254.2005.00158.x
  5. Zhai, X.-X.; Meng, X.-H.; Wang, C.-B.; Zhao, Y.-M.; Yang, J.-L., Anti-hypoxic    active constituents from the twigs of Zanthoxylum armatum DC. and their chemotaxonomic significance. Biochemical Systematics and Ecology 2022, 104, 104480.doi:10.1016/j.bse.2022.104480

Round 2

Reviewer 1 Report

All the comments mentioned by the reviewer were answered and modifications were made in the manuscript. I'm pleased with your corrections. Therefore, I recommended this work can be accepted. 

Author Response

Dear reviewer,

Thanks very much for your decision and appreciate your comments. We wish to express our thankfulness for the kind remarks and positive comments on our paper.

Point 1: All the comments mentioned by the reviewer were answered and modifications were made in the manuscript. I'm pleased with your corrections. Therefore, I recommended this work can be accepted.

Response 1: Dear reviewer, thanks for your patience and suggestions for improvement.

Reviewer 3 Report

In the current version of the manuscript the authors addressed most of the previous suggestions. However, there are still some issues that need be addressed before publication. Also, I encourage the authors to be careful when making changes in order to avoid to introduce new mistakes.

Major changes.

Point 1: The authors must significantly improve the writing of the manuscript. The current version was improved, but the manuscript still needs to be completely and carefully revised. Some examples are shown:

Paragraph: The relative expression of IκBα in Figure 4C showed that the degradation of IκBα protein was not inhibited in TJZ-1 (1 and 10 μM) and TJZ-2 (1 and 10 μM) treatment. Moreover, the phosphorylation of p65 were significantly inhibited in TJZ-1 (10 μM) and TJZ-2 (10 μM) treatment (Figure 4D). These results suggested that TJZ-1 and TJZ-2 inhibit LPS-induced HMC3 activation via the NF-κB signaling pathway. Especially, TJZ-1 and TJZ-2 inhibited the activation of IKK complex by other potential means.

Suggestion: The relative expression of IκBα in Figure 4C showed that the degradation of IκBα protein was not inhibited by treatment with TJZ-1 (1 and 10 μM) and TJZ-2 (1 and 10 μM). Moreover, p65 phosphorylation was significantly inhibited by treatment with 10 μM of TJZ-1 and TJZ-2, respectively (Figure 4B). These results suggested that TJZ-1 and TJZ-2 inhibit LPS-induced HMC3 activation via the NF-κB signaling pathway.

Moreover, it is unclear what the author mean whit “Especially, TJZ-1 and TJZ-2 inhibited the activation of IKK complex by other potential means”

Section 2.4. The description of results of figure 4 have several mistakes. For example in the paragraph we can read “As shown in Figure 4B, TJZ-1 (1 and 10 μM) significantly inhibited the phosphorylation of IKK.” However, the results of Ikk are in figure 4D.

Another example is “Moreover, the phosphorylation of p65 were significantly inhibited in TJZ-1 (10 μM) and TJZ-2 (10 μM) treatment (Figure 4D)” But the results of p65 are in panel 4b,

Please correct it and describe the results of each panel in the order they appear in the figure.

Author Response

Dear reviewer,

Thank you so much for handling the review of our manuscript. We appreciate the promoting comments to our study, and we have accepted and revised as recommended in this revised manuscript. The point-by-point responses are provided below. We hope that the revision is acceptable, and your favorable consideration of our manuscript is greatly appreciated.

Point 1: The authors must significantly improve the writing of the manuscript. The current version was improved, but the manuscript still needs to be completely and carefully revised. Some examples are shown:

Paragraph: The relative expression of IκBα in Figure 4C showed that the degradation of IκBα protein was not inhibited in TJZ-1 (1 and 10 μM) and TJZ-2 (1 and 10 μM) treatment. Moreover, the phosphorylation of p65 were significantly inhibited in TJZ-1 (10 μM) and TJZ-2 (10 μM) treatment (Figure 4D). These results suggested that TJZ-1 and TJZ-2 inhibit LPS-induced HMC3 activation via the NF-κB signaling pathway. Especially, TJZ-1 and TJZ-2 inhibited the activation of IKK complex by other potential means.

Suggestion: The relative expression of IκBα in Figure 4C showed that the degradation of IκBα protein was not inhibited by treatment with TJZ-1 (1 and 10 μM) and TJZ-2 (1 and 10 μM). Moreover, p65 phosphorylation was significantly inhibited by treatment with 10 μM of TJZ-1 and TJZ-2, respectively (Figure 4B). These results suggested that TJZ-1 and TJZ-2 inhibit LPS-induced HMC3 activation via the NF-κB signaling pathway.

Response 1: According to the comment, the manuscript has been revised.

Point 2: Moreover, it is unclear what the author mean whit “Especially, TJZ-1 and TJZ-2 inhibited the activation of IKK complex by other potential means”.

Section 2.4. The description of results of figure 4 have several mistakes. For example in the paragraph we can read “As shown in Figure 4B, TJZ-1 (1 and 10 μM) significantly inhibited the phosphorylation of IKK.” However, the results of Ikk are in figure 4D.

Please correct it and describe the results of each panel in the order they appear in the figure.

Response 2: According to the comment, the manuscript has been revised.
